# The Natural Tendency of Feed Forward Neural Networks to Favor Invariant Units

**Dean A. Pospisil  and Wyeth Bair**
Department of Biological Structure
University of Washington
Seattle, WA 98103
deanp3@uw.edu

## Abstract

A central goal in the study of the primate visual cortex and hierarchical models for object recognition is understanding how and why single units trade off invariance versus sensitivity to image transformations. For example, in both deep networks and visual cortex there is substantial variation from layer-to-layer and unit-to-unit in the degree of translation invariance. Here, we provide theoretical insight into this variation and its consequences for encoding in a deep network. Our critical insight comes from the fact that rectification simultaneously decreases response variance and correlation across responses to transformed stimuli, naturally inducing a positive relationship between invariance and dynamic range. Invariant input units then tend to drive the network more than those sensitive to small image transformations. We discuss consequences of this relationship for AI: deep nets naturally weight invariant units over sensitive units, and this can be strengthened with training, perhaps contributing to generalization performance. Our results predict a signature relationship between invariance and dynamic range that can now be tested in future neurophysiological studies.

## 1 Introduction

Invariances to image transformations, such as translation and scaling, have been reported in single units in visual cortex, but just as often sensitivity to these transformations has been found (El-Shamayleh and Pasupathy, 2016, Sharpee et al. 2013, Rust and DiCarlo, 2012). Similarly, in deep networks there is variation in translation invariance both within and across layers (Pospisil et al., 2018, Shen et al., 2016, Shang et al., 2016, Goodfellow et al., 2009). Notionally, information about the position of the features composing objects may be important to category selectivity. For example, the detection of eyes, nose, and lips are not sufficient for face recognition, the relative positions of these parts must also be encoded. Thus it is reasonable to expect some balance between invariance and sensitivity to position. We empirically observe that in a popular deep network, in both its trained and untrained state, invariant units tend to have higher dynamic range than sensitive units (Figure 1B and C). This raises the possibility that the effective gain on invariant units into the subsequent layer is stronger than that of sensitive units. Here we provide theoretical insight into how rectification in a deep network could naturally bias networks to this difference between invariant and sensitive units. We do this by examining how co-variance of a multivariate normal distribution is influenced by rectification, and we then test these insights in a deep neural network.

## 2 Statistical model

The response of a unit in a feed forward neural network is: $r = \vec{w} \cdot g(S)$ where $S$ is the response of all $n$ input units in the previous layer, $g$ the non-linearity of rectification $g(x) = max(0, x)$, $\vec{w}$ is the $n \times 1$ vector of weights, and $r$ is the response of the unit. Randomly sampling from a distribution of input images, the response $S$ takes on a distribution with some expectation and covariance across these images: $E[S] = \mu$ (an $n \times 1$ vector), and $Cov[S] = \Sigma$ (an $n \times n$ matrix). The application of the non-linearity transforms these moments: $E[g(S)] = \tilde{\mu}$, $Cov[g(S)] = \tilde{\Sigma}$. Let $S_1$ be the responses to randomly sampled input images and $S_2$ the responses to a transformation of those same images. So the moments of the full distribution are:

$$E\begin{bmatrix} \tilde{S}_1 \\ \tilde{S}_2 \end{bmatrix} = \begin{bmatrix} \tilde{\mu}_1 \\ \tilde{\mu}_2 \end{bmatrix}, \quad \text{Cov}\begin{bmatrix} \tilde{S}_1 \\ \tilde{S}_2 \end{bmatrix} = \begin{bmatrix} \tilde{\Sigma}_{1,1} & \tilde{\Sigma}_{1,2} \\ \tilde{\Sigma}_{2,1} & \tilde{\Sigma}_{2,2} \end{bmatrix},$$

where $\tilde{\Sigma}_{1,1}$ is the covariance of rectified input units responding to the original images, $\tilde{\Sigma}_{2,2}$ is the covariance of rectified input units to the transformed images and $\tilde{\Sigma}_{1,2}$ is the covariance between rectified input units responding to the reference and transformed

33rd Conference on Neural Information Processing Systems (NeurIPS 2019), Vancouver, Canada.

images. We note we only define the 1st two moments above and no assumption about the distribution of the *rectified* responses is made. The covariance of an output unit with weights $\vec{w}$ on the $n$ rectified input units is:

$$\begin{bmatrix} \tilde{\sigma}_1^2{}' = \vec{w}\tilde{\Sigma}_{1,1}\vec{w}^T & \tilde{\rho}'\tilde{\sigma}_1'\tilde{\sigma}_2' = \vec{w}\tilde{\Sigma}_{1,2}\vec{w}^T \\ \tilde{\rho}'\tilde{\sigma}_1'\tilde{\sigma}_2' = \vec{w}\tilde{\Sigma}_{2,1}\vec{w}^T & \tilde{\sigma}_2^2{}' = \vec{w}\tilde{\Sigma}_{2,2}\vec{w}^T \end{bmatrix}$$

so the correlation between the response of the output unit to the reference and transformed images is:

$$\tilde{\rho}' = \frac{\vec{w}\tilde{\Sigma}_{1,2}\vec{w}^T}{\sqrt{\vec{w}\tilde{\Sigma}_{1,1}\vec{w}^T\,\vec{w}\tilde{\Sigma}_{2,2}\vec{w}^T}}.$$

Below we investigate how the $\tilde{\Sigma}_{i,j}$ depend on $\Sigma_{i,j}$ and $\mu$, which provides insight into a relationship between $\tilde{\rho}'$ and $\tilde{\sigma}^2{}'$. We begin by examining a model of a single rectified input unit responding to the reference and transformed images.

## 2.1 Single rectified input

We model the responses, $S_1$ and $S_2$, of a single input unit to the reference and transformed inputs, respectively, as a bivariate normal distribution:

$$\begin{bmatrix} S_1 \\ S_2 \end{bmatrix} \sim N\left( \begin{bmatrix} \mu \\ \mu \end{bmatrix}, \begin{bmatrix} \sigma^2 & \rho\sigma^2 \\ \rho\sigma^2 & \sigma^2 \end{bmatrix} \right).$$

When these responses are acted on by rectification, both the variances of the responses and the correlation between the sets of responses is decreased. This observation is analogous to that of de la Rocha et al. (2007) where they investigated the influence of neuronal firing threshold rectification on the pairwise correlations between neurons as a function of firing rate. We extend this observation in the next section to consider how this effect influences invariance in downstream units.

It is instructive to consider a schematic (Figure 2A) of the distribution of responses. The fall in correlation occurs because the variance from the linear relationship $Var(E[\tilde{S}_2|\tilde{S}_1])$ decreases as a result of truncation, whereas the residual $E[Var(\tilde{S}_1|\tilde{S}_1)]$ is not reduced as much. (see Intuition behind variance correlation relation). For further demonstration, see de la Rocha et al. (2007).

In the following section, we will write the correlation and variance after rectification explicitly as a function of the relevant parameters: $\tilde{\sigma}(\mu/\sigma)^2$ and $\tilde{\rho}(\mu/\sigma, \rho)$.

## 2.2 Unit integrating across multiple rectified input units

Here we extend from the single input unit case to the multi-input unit case by examining the invariance $\tilde{\rho}'$ resulting from taking weighted combinations of rectified input units. Our key insight is that invariance increases to the degree that directions of maximal variance in the response distribution of rectified input units are integrated. For a first order approximation of this relationship we approximate input covariance before rectification as an identity matrix scaled by the average variance:

$$\Sigma = \text{diag}(\Sigma) + (\Sigma - \text{diag}(\Sigma)) \approx \text{diag}(\Sigma) \approx I\frac{1}{n}\sum_{i=1}^{n}\sigma_{i,i}^2 = I\bar{\sigma}^2$$

This approximation improves as off diagonal covariance shrinks and $\frac{\bar{\sigma}^2}{Var(\sigma_i^2)}$ increases. Thus our approximate model is: $Cov[S_1] = Cov[S_2] = \Sigma_{1,1} = \Sigma_{2,2} = \bar{\sigma}^2 I$ where $\bar{\sigma}^2$ is averaged across the diagonals of the original $S_1$ and $S_2$ $E[S_1] = E[S_2] = \mu = [\mu_1...\mu_n]^T$ where means are approximated by averaging across the original $S_1$ and $S_2$. $Cov[S_1, S_2] = \Sigma_{1,2} = \rho\bar{\sigma}^2 I$ where $\rho\bar{\sigma}^2 > 0$ justified by the assumption of a small transformation thus correlation is positive.

For convenience sort the $\mu_i$ in from high to low, then it naturally follows that the eigenvectors of $\tilde{\Sigma}_{1,2}$ and $\tilde{\Sigma}_{1,1}$ are the same: $I$ the identity (since the covariance matrices are diagonal) and the eigenvalues are simply the entries of the diagonal (in order from high to low since $\tilde{\rho}$, and $\tilde{\sigma}^2$ are decreasing in $\mu$; see Figure 2B) so we have:

$$\tilde{\rho}' = \frac{\sum_i^n w_i^2\tilde{\sigma}^2(\mu_i/\sigma)\tilde{\rho}(\mu_i/\sigma,\ \rho)}{\sum_{i=1}^n w_i^2\tilde{\sigma}^2(\mu_i/\sigma)}$$

thus we have the geometric picture described in Figure 3A exactly. The denominator as a function of the direction of a unit length $\vec{w}$ (length of $\vec{w}$ does not change $\rho'$) is an axis aligned ellipsoid with length along the $i$th axis of $\tilde{\sigma}^2(\mu_i/\sigma)$. Notice that the numerator is the variation of the output unit thus more invariant units contribute more variance than less invariant units assuming there is not a negative correlation between $w_i^2$ and $\tilde{\sigma}_i^2\tilde{\rho}_i$. The numerator is another axis aligned ellipsoid (blue) with length along the $i$th axis of $\tilde{\sigma}^2(\mu_i/\sigma)\tilde{\rho}(\mu_i/\sigma,\ \rho)$ this numerator ellipsoid is contained within the denominator since $\tilde{\rho}' \leq 1$. Recognizing $\tilde{\rho}'$ as a weighted arithmetic mean with with weights $c_i = \frac{w_i^2\tilde{\sigma}^2(\mu_i/\sigma)}{\sum_{i=1}^n w_i^2\tilde{\sigma}^2(\mu_i/\sigma)}$ (note $\sum c_i = 1$) we see that if there is not a negative correlation of $w_i^2$ with $\tilde{\sigma}_i^2$ than output units will tend to have a higher invariance then the average of input units thus pushing output units invariance upwards.

Performing simulations of a few simple input unit covariance structures shows that the $\tilde{\rho}'$ to $\tilde{\sigma}^2{}'$ relationship is maintained, though its form changes (Figure 3B). Integrating over a population of input units the form of the relationship changes from the single input unit case (Figure 3B black dashed and dotted line). Averaging over input units with different $\mu_i/\sigma$ shifts the relationship because of the curvature of the non-linear transform of the ratio $\tilde{\rho}(\mu/\sigma, \rho)$. Finally, in the case where input units have non-zero correlation (i.e. shared tuning, off-diagonals of $\Sigma_{i,j}$ non-zero; Figure 3B red and cyan) overall variance increases because correlated input units are being added.

## 3   Experimental results

Here we analyze the covariance structure of the inputs of a popular deep neural network (AlexNet) for translations of input images. We first tested the network in its *untrained* state by presenting a collection of 500 image patches drawn from the 2012 ImageNet validation set. References images were cropped enough to allow the original and translated images to fit within the maximal receptive field of the units being tested. We included a small and a large translation, and at each convolutional layer we measured the correlation and variance. We find a positive relationship at all layers with significant Spearman's ranked correlation for both transformations (Table 1 Untrained). The strength of the relationship tended to be stronger for the smaller translation (Figure 1C, orange). Thus in a popular deep network with no training, units which tended to have greater invariance also had higher dynamic range. We repeated the same analysis in AlexNet after it was fully trained for object recognition. Again we observed a significant positive relationship (Table 1 Trained; Figure 1B). The relationship was somewhat weaker than in the trained network. Thus training weakens but does not remove the bias of the network to associate higher dynamic range with higher translation invariance.

| Layer | Distance | Untrained | Trained |
|-------|----------|-----------|---------|
| Conv2 | Far      | 0.44      | 0.35    |
|       | Close    | 0.92      | 0.58    |
| Conv3 | Far      | 0.67      | 0.56    |
|       | Close    | 0.94      | 0.75    |
| Conv4 | Far      | 0.65      | 0.48    |
|       | Close    | 0.92      | 0.65    |
| Conv5 | Far      | 0.64      | 0.18    |
|       | Close    | 0.86      | 0.64    |

Table 1: Table of Spearman's ranked correlation coefficient ($r_s$) between $\rho$ and $\sigma^2$ in AlexNet layers and for the trained and untrained network. All values $p < 0.001$.

Finally, we asked whether the network may compensate for this imbalance by placing weights of higher magnitude on low dynamic range units (a negative correlation between $\tilde{\sigma}_i^2$ and $w_i^2$), thus effectively removing this bias. We measured whether the percent of weight magnitude on a given input unit across output units was greater for input units with higher variance. We found Conv3 ($r_s = 0.34$) and Conv4 ($r_s = 0.19$) tended to have higher weights on higher variance input units while there was no correlation in Conv2 and and Conv5. This indicates the network does not compensate for the imbalance in dynamic range between invariant and sensitive units but actually sometimes emphasizes it.

## 4   Discussion

We have documented an empirical relationship between the dynamic range of unrectified units in a deep network and their invariance. We provided a simple 1st order statistical model to explain this effect in which rectification caused the population representation to primarily vary in dimensions that were invariant to small image perturbations, whereas small perturbations were represented in directions of lower variance. Further work can investigate whether this imbalance improves generalization because of the emphasis placed on invariant over sensitive units.

We note this relationship is weaker in the trained then untrained network further work can udnerstan this difference. Our approximations assumed low covariance between input units and homoegenous input variance while this may be expected in a random network it may not be true in a trained network. More crucially further theoretical work should consider the influence of co-variance between input units and invariance of output units as a function of weights.

To extend insights from simplified, artificial networks to neurobiology, it will first of all be important to test whether cortical neurons showing more invariance also tend to have a higher dynamic range. If they do, this will establish a fundamental theoretical connection between computations of deep networks and the brain.

## 5   Response to reviewer comments

We thank our reviewers for their careful and insightul comments. Above we have taken their comments into account in editing our final draft. Below we address their three main concerns.

### 5.1   Intuition behind variance correlation relation

It is instructive to consider a schematic (Figure 2A) of the distribution of responses. The probability mass of the response is broken into 4 quadrants, the 1st green is unaffected by rectifications, the 2nd (purple) is projected onto the vertical axis (thick purple), the 3rd (red) is projected onto the origin (red dot), and the 4th (green) projected onto the horizontal axis (thick green). The diagonal line is the line of best fit expressing the linear relationship and the vertical line is a conditional distribution whose variance is the conditional residual which averaged gives the residual variance of the linear relationship. $\tilde{\sigma}^2$ decreases as $\mu/\sigma$ decreases because the spread of the distribution is truncated to the degree that the distribution falls beneath the threshold. For correlation it is useful

to consider:

$$\tilde{\rho}^2 = \frac{\text{Var(Best Linear Predictor)}}{\text{Var(Best Linear Predictor)} + \text{Var(Residual)}} = \frac{Var[E[\tilde{S}_2|\tilde{S}_1]]}{Var[E[\tilde{S}_2|\tilde{S}_1]] + E[Var(\tilde{S}_2|\tilde{S}_1)]}$$

where $Var[E[\tilde{S}_2|\tilde{S}_1]] = Var[\tilde{\rho}\tilde{S}_1] = \rho^2 Var(\tilde{S}_1)$ decreases more rapidly then the average residual $E[Var(\tilde{S}_2|\tilde{S}_1)]$. Notionally we can think of $Var[E[\tilde{S}_2|\tilde{S}_1]]$ as the vertical height of the diagonal line that has not been truncated (solid not truncated, dashed truncated) in Figure 2A and $E[Var(\tilde{S}_2|\tilde{S}_1)]$ as the average length of vertical lines not truncated. Notice that the ratio of truncated to untruncated is lower for the diagonal then the vertical average. In the figure at $\mu = 0$ the diagonal line is cut in half and so is the length of a vertical line drawn here. But at all other positions ($\mu > 0$) where a vertical line would be drawn the vertical solid lines length is truncated less than half thus on average the vertical line is less truncated than the diagonals vertical length.

### 5.2 Rectification does not explain invariance

We would like to emphasize we are not arguing that rectification explains the generalization properties of networks only that its influence on covariance may be one of many factors influencing invariance.

### 5.3 Justification of simplifying assumptions

We would like to emphasize that in this paper we pursue intuition by trying to understand a simple approximation to rectifications influence on invariance which results in a simple analytic form. Our first approximation is to remove off-diagonal covariances. Since the influences of off-diagonals are additive they can be seen as modulating the effects induced by the diagonals:

$$\vec{w}^T \tilde{\Sigma} \vec{w} = \vec{w}^T \text{diag}(\tilde{\Sigma}) \vec{w} + \vec{w}^T (\tilde{\Sigma} - \text{diag}(\tilde{\Sigma}) \vec{w} \approx \vec{w}^T \text{diag}(\tilde{\Sigma}) \vec{w}$$

Thus here we analytically study the first order effect of rectification in output neurons on the basis of the variance but not covariance of their inputs.

Finally we approximate the diagonal of the input variance with the average variance an approximation which minimizes squared error

$$\vec{w}^T \text{diag}(\Sigma) \vec{w} \approx \vec{w}^T I \frac{1}{n} \sum_{i=1}^{n} \sigma_{i,i}^2 \vec{w} = \vec{w}^T I \bar{\sigma^2} \vec{w}$$

variation in $\sigma^2$ will hurt the strength of this approximation but not change the main effect unless this variation is negatively correlated with $\mu_i$ thus canceling out the relationship between correlation and variance. We would not expect this negative correlation in an untrained network and further work can check whether this, potentially interesting, relationship exists in trained networks. We note that normalization enforces this approximation and thus these approximations may be particularly suited to networks using normalization.

## Acknowledgements

This work was funded by a National Science Foundation (NSF) Graduate Research Fellowship (D.A.P.), the National Science Foundation CRCNS Grant IIS-1309725 (W.B.), and NIH Grant R01 EY-027023 (W.B.)

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

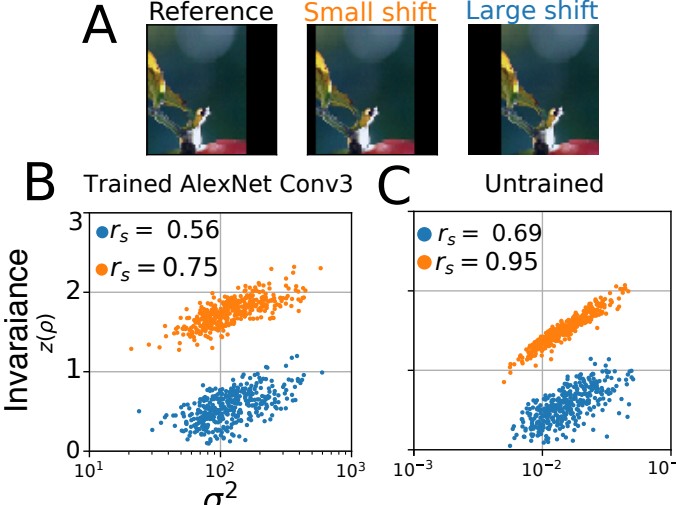

**Figure 1:** Empirical relationship between invariance and response variance. **(A)** Example stimuli conditions: reference position images cropped to 3/4 of unit maximal receptive field, then consider the correlation of responses across images for a small shift (1/8 of RF) and large shift (1/4 of RF). **(B)** The variance of unrectified responses from Conv3 units in trained AlexNet to the reference position plotted against Fisher's z of reference images and shift images (orange small shift, blue large shift). Both show strong positive relationship with Spearman's ranked correlation coefficient ($r_s$) listed in legend. **(C)** Same plot for the untrained AlexNet showing little change in the form of the relationship except for its strength.

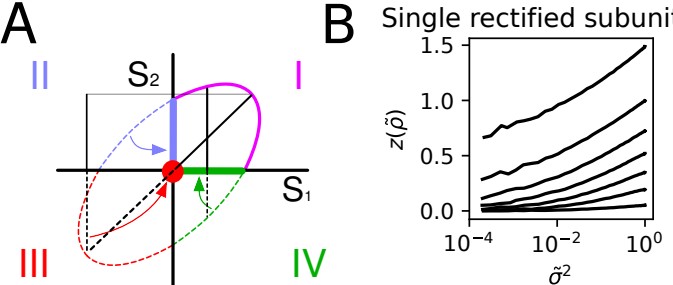

**Figure 2:** Influence of rectification on variance and correlation in bivariate normal. **(A)** Transformation of bivariate normal by rectification. Distribution in quadrant I is preserved (pink), quadrant II, IV collapsed onto $S_1$ and $S_2$ axis respectively (thick green, blue lines), and III mapped onto origin (red). **(B)** Plotting $\tilde{\sigma}^2(\mu/\sigma)$ against $\tilde{\rho}(\mu/\sigma, \rho)$ there is a positive relationship because both are increasing with $\mu/\sigma$. $\tilde{\rho}$ is transformed to Fisher's z and $\tilde{\sigma}^2$ plotted on log axis revealing an approximate relationship: $a(\tilde{\sigma}^2)^b = z(\tilde{\rho})$.

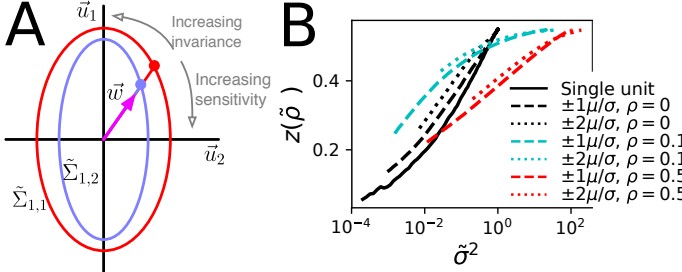

**Figure 3:** $\tilde{\rho}'$ to $\tilde{\sigma}^{2}{}'$ relationship in unrectified unit resulting from integrating over rectified input units. **(A)** Schematic intuition of $\tilde{\rho}'$ to $\tilde{\sigma}^{2}{}'$ relationship. In blue $\tilde{\Sigma}_{1,2}$, in red $\tilde{\Sigma}_{1,1}$, pink vector is $\vec{w}$. Axes are eigenvectors with vertical axis having larger associated eigenvalue. Ratio of $\vec{w}^T \tilde{\Sigma}_{1,2} \vec{w}$ and $\vec{w}^T \tilde{\Sigma}_{1,1} \vec{w}$ increases as $\vec{w}$ points in direction of greater variation of both. **(B)** Simulation of integrating over populations of rectified input units varying average $\mu/\sigma$ (within a trace), input unit correlation ($\rho$ across colors), and neighborhood size of $\mu/\sigma$ averaged over (line pattern).

