# OpenReview forum: "The Natural Tendency of Feed Forward Neural Networks to Favor Invariant Units"
_NeurIPS.cc/2019/Workshop/Neuro_AI — Real Neurons & Hidden Units @ NeurIPS 2019 Poster_

### Official Review · AnonReviewer3 · 2019-09-25
**Can rectification nonlinearity explain translational invariance?**

**Clarity:** 1

**Comment:**

The idea behind the paper is appealing, but more effort needs to be put into clarifying the reasonings. An intuitive explanation of the suggested relationship between variance and translational invariance can help. Also, the statistical deserve a better justification.
The figures were helpful, but more needs to be said on their details, e.g. Figures 2B and 3B.


**Category:**

Common question to both AI & Neuro

**Clarity Comment:**

It was quite difficult to follow the line of reasoning in the paper. The connection between the subsections was not clear enough; e.g. it was difficult (or maybe impossible) to understand how the materials presented in section 2.1 were used in the arguments in section 2.2. While reading the paper, it felt that there should be a simple intuition behind the hypothesis (the relationship between translation invariance and the variance of units), though without sufficient explanation, it was hard to grasp that.
Also, in different parts of the paper, units and subunits' statistics were discussed interchangeably. It was almost impossible to distinguish these two, and it seemed that understanding the differences in the notations was key to comprehending the idea.

**Evaluation:**

2: Poor

**Importance:**

3: Important

**Importance Comment:**

Invariance to translation and other affine transformations is necessary for object recognition. In both vision neuroscience and deep learning the problem of invariance has been discussed extensively, yet there is no consensus on the computational basis of it in the brain and deep nets. This paper suggests that the rectification nonlinearity in deep networks underlies the emergence of invariance in these models.

**Intersection:**

4: High

**Intersection Comment:**

The question is important to both neuroscience and AI. The hypothesis suggested in the paper could be applicable to both fields.

**Rigor Comment:**

There are many simplifying assumptions which, even if reasonable, are hard to accept based on their presentations in the paper: a very simple one is the assumption of Gaussian distribution for both S and g(S). The authors certainly know that if S has a Gaussian distribution, g(S) doesn't necessarily follow the same distribution. For g(x) = max(0,x), g(S) would follow the same distribution, only for large mean values. In that case, you can simply assume that the nonlinearity has almost no effect on the distribution which is against the important role that the paper is attributing to the nonlinearity.

Moreover, it seems that some of the simplifying assumptions has made the problem of invariance too simple; that is, the final conclusion is the effect of simplifying assumptions in the process: as an example, \Sigma_{1,1} = \Sigma_{2,2} = e*\Sigma_{1,2}. Based on this assumption, the second-order statistics of subunits do not change much with translation which simplifies the main problem significantly.
An initial intuitive explanation of the idea behind the hypothesis in the paper would extremely help for following the mathematical arguments later on in the paper.

**Technical Rigor:**

1: Not convincing

---

> ### Comment · ~Dean_A_Pospisil1 · 2019-10-03
> **Response to reviewers main comments.**
>
> Reviewer 1’s comments overlap with reviewer 2’s here we emphasize comments unique to reviewer 2. We put sections from reviewers comments in quotes.
>
> ‘Title: Can rectification nonlinearity explain translational invariance?’
>
> We will further clarify our paper to emphasize it is not arguing rectification explains translation invariance only that it causes invariant units in the network to also have a higher dynamic range thus have a potentially increased influence on the responses of downstream units assuming network weights do not compensate for this imbalance. We provide evidence the network does not compensate in Results paragraph 2.
>
> ‘The authors certainly know that if S has a Gaussian distribution, g(S) doesn't necessarily follow the same distribution.’
>
> We do know this and will make sure it is clear that we do not assume g(S) is gaussian distributed only that its first two moments exist (section 2 first stand alone equation).
>
> ‘An initial intuitive explanation of the idea behind the hypothesis in the paper would extremely help for following the mathematical arguments later on in the paper.’
>
> We will emphasize intuition early in our second draft. In brief: the more of a subunits response is rectified the less variance it will have and also the less correlation it will have (in general and across translations) thus more invariant units will have higher variation (dynamic range).

---

### Official Review · AnonReviewer1 · 2019-09-25
**Potentially interesting work, but difficult to read and with unclear significance**

**Clarity:** 2

**Comment:**

This submission proposes suggests that neurons with higher variance tend to be more invariant to positional shifts because of properties of the ReLU activation, and that neural networks may thus naturally weight invariant units over less invariant units. The idea of investigating neural networks through this lens is potentially interesting, and the authors perform experiments on activations extracted from AlexNet to validate their hypothesis. However, much of this submission is difficult to follow, and in its current form, it is neither mathematically precise nor intuitively easy to grasp. Several simplifying assumptions are made in order to derive the conclusions, and it's not clear why these assumptions should be plausible or whether they approximate the empirical behavior of real neural networks.

**Category:**

Neuro->AI

**Clarity Comment:**

There are many issues with the clarity of this work.

1. Section 2 starts out by talking about "subunits" in the previous layer. This is not standard terminology for artifical NNs and it's unclear what it refers to. S_1, S_2 are introduced but then \tilde S_1 and \tilde S_2 are used in the equation below. I think \bar S_1 is g(S_1) but I'm not sure.
2. In Section 2.1, S_1 and S_2 seem to go from being matrices to vectors and there seems to be an implicit assumption at S_1 and S_2 are identically distributed.
3. Although Figure 2A is clear, the actual relationship between variance and correlation is never described mathematically. Instead, the authors point to de la Rocha (2007).
4. Section 2.2 is hard to understand for many reasons. It's not clear the stated assumption that "each subunit has the same covariance structure" means mathematically. The assumption on the eigenvectors is written as an equation involving \mu but the equation above seems to use u. I don't see how to get Figure 3A from the equation given.

**Evaluation:**

2: Poor

**Importance:**

2: Marginally important

**Importance Comment:**

Investigating how representational properties (invariance) might relate to generalization properties seems like an appealing research direction. However, I'm unconvinced that this particular approach to understanding this connection, via analyzing what happens when ReLU is applied to Gaussian-distributed data, is particularly informative. Additionally, the writing is unclear in many places and many assumptions are made with little theoretical or empirical justification.

**Intersection:**

4: High

**Intersection Comment:**

This work relates previous ideas regarding correlation between spike trains to properties of deep neural networks. Generally, the idea of linking representational properties (typically a stronger focus in neuroscience than ML) to inherent generalization properties of systems (of interest to both ML and neuroscience, but easier to study in ML) is an interesting area at the intersection between AI and neuroscience.

**Rigor Comment:**

It's challenging to verify this work, since many assumptions are introduced and a lot of math seems to have been omitted for brevity, including a derivation of the "critical insight" that "rectification simultaneously decreases response variance and correlation across responses to transformed stimuli." But even if one were to assume that the conclusions are correct as stated, several assumptions in Section 2.2 are not a priori plausible to me and have neither heuristic nor empirical justification. It's not clear to me why we should assume that $\Sigma_{1,1} = \Sigma_{2, 2} = \Sigma_{1,2}\epsilon$, that "each subunit has the same covariance structure as another" (or what this means mathematically), or that the eigenvectors are the same. Each of these assumptions seems quite strong. It's intuitively implausible that ReLU activation alone can explain the generalization properties of neural nets without any further assumptions, so these assumptions seem essential to the argument, but the submission provides no theoretical or empirical justification.

The experimental results are somewhat more convincing, although the statement in the abstract "deep nets naturally weight invariant units over sensitive units, and this can be strengthened with training" seems to be somewhat contradicted by the experimental results, which show that the described effect is much more prominent at initialization than in trained networks.

**Technical Rigor:**

2: Marginally convincing

---

> ### Comment · ~Dean_A_Pospisil1 · 2019-10-03
> **Response to reviewers main conceptual comments.**
>
> Below are direct responses to the reviewers careful and helpful comments. We condense the reviewers comment (we retain original wording so original comments can be found) and focus on conceptual issues over clarity issues to fit under the character limit.
>
> ‘I'm unconvinced that this particular approach to understanding this connection, via analyzing what happens when ReLU is applied to Gaussian-distributed data, is particularly informative.’
>
> We agree the work will be more convincing if  we show that units variance and correlation follow the gaussian predictions. We will include predictions from the gaussian case overlaid on the measured relationship in the network. We will make it clearer in the next draft that the form of the variance-correlation relationship which we show in gaussian simulations (Figure 3B) is found in the deep neural network (Figure 1B) providing evidence of that the gaussian case is informative.
>
> ‘derivation of the "critical insight" that "rectification simultaneously decreases response variance and correlation across responses to transformed stimuli.”’
>
> We cite de la Rocha 2007 that originally provided intuition for this insight in this case of between neuron correlations (in their paper see Figure 4D correlation goes down as more of PDF falls beneath rectification threshold as does marginal variance), but we agree it would be helpful to have a figure in our own paper developing this insight in addition we have developed analytic approximations for the effect that provide further insight.
>
> ‘ why we should assume that $\Sigma_{1,1} = \Sigma_{2, 2} = \Sigma_{1,2}\epsilon$,’
>
> This assumption can be justified from the deep network being a piecewise continuous function of affine transformations of its inputs in this case translation. Basically in the case where there is no translation then $\Sigma_{1,1} = \Sigma_{2, 2} = \Sigma_{1,2}$ holds then we can achieve any given level of approximation of this formula via a small enough translation. We will include this justification in addition we can provide measurements in the network that show the approximation is good even for relatively large translations.
>
> ‘It's intuitively implausible that ReLU activation alone can explain the generalization properties of neural nets’
>
> We will make it clearer that we are not arguing ‘ReLU activation alone can explain the generalization properties of neural nets’. Many other properties are involved in developing invariance. Our point is that units invariant to small perturbations will tend have a higher dynamic range and this will give them an outsized influence on the response of downstream units assuming the network does not  learn weights which cancel out this effect. We provide evidence that the network does not ‘cancel’ out this tendency in the finding that weights on high variance subunits does not tend to be lower, in fact it is higher in some layers (Experimental results, 2nd paragraph)
>
> ‘"deep nets naturally weight invariant units over sensitive units, and this can be strengthened with training" seems to be somewhat contradicted by the experimental results, which show that the described effect is much more prominent at initialization than in trained networks.’
>
> ‘We will make it clearer that what we mean by "deep nets naturally weight invariant units over sensitive units and this can be strengthened with training” is that in their untrained state this variance-correlation relationship is strong thus it happens ‘naturally’ without any training signals. The word ‘strengthened’  was too ambiguous: we were referring to the fact that the network places greater weight on units with a higher dynamic range. We agree the strength of the relationship is weaker, it is the form (i.e. slope) of the relationship that can be strengthened with training.’
>
> ‘the actual relationship between variance and correlation is never described mathematically. Instead, the authors point to de la Rocha (2007).’
>
> We appreciate this criticism as the relationship between variance and correlation  has yet to be described in an analytic form de la Rocha provide simulations and intuition. Space permitting we will include some novel insightful approximations to the relationship we have derived.
>
> ‘I don't see how to get Figure 3A from the equation given.’
>
> We will further clarify the ellipsoids describe the variance of resulting from $w^T \Sigma w$ as a function of a fixed length w for $\Sigma_{1,1}$ (outer ellipsoid red) and $\Sigma_{1,2}$ inner ellipsoid purple. In simulation we show (FIG 3b) that when $w$ points in directions of higher variance it monotonically gives higher invariance as in (FIg 3A). Space permitting we will provide a toy model that makes the connection between Fig 3a and the equation given clear and of a simple analytic form.
>
> We thank the reviewer for their careful reading and detailed comments which will make our conclusions clearer and hopefully more compelling. We will include more intuition

---

### Decision · Program_Chairs · 2019-10-02

Accept (Poster)